# Neurohabilitation of Cognitive Functions in Pediatric Epilepsy Patients through LEGO^®^-Based Therapy

**DOI:** 10.3390/brainsci14070702

**Published:** 2024-07-13

**Authors:** Flor Lorena Zaldumbide-Alcocer, Norma Angélica Labra-Ruiz, Abril Astrid Carbó-Godinez, Matilde Ruíz-García, Julieta Griselda Mendoza-Torreblanca, Lizbeth Naranjo-Albarrán, Noemí Cárdenas-Rodríguez, Erika Valenzuela-Alarcón, Eduardo Espinosa-Garamendi

**Affiliations:** 1Servicio de Neurología, Dirección Médica, Instituto Nacional de Pediatría, Mexico City 04530, Mexico; florlorenaza@gmail.com (F.L.Z.-A.); matilderuiz@gmail.com (M.R.-G.); 2Laboratorio de Neurociencias, Subdirección de Medicina Experimental, Instituto Nacional de Pediatría, Mexico City 04530, Mexico; norma_labra@yahoo.com.mx (N.A.L.-R.); julietamt14@hotmail.com (J.G.M.-T.); noemicr2001@yahoo.com.mx (N.C.-R.); 3Unidad de Neurohabilitación y Conducta, Subdirección de Medicina, Dirección Médica, Instituto Nacional de Pediatría, Mexico City 04530, Mexico; abi.astrid@hotmail.com; 4Fundación COGNITIVE HABILITATION, Mexico City 03100, Mexico; 5Departamento de Matemáticas, Facultad de Ciencias, Universidad Nacional Autónoma de México, Mexico City 04510, Mexico; lizbethna@ciencias.unam.mx; 6Fundación Care & Share for Education, Mexico City 01050, Mexico; erika@edacom.mx

**Keywords:** cognitive functions, LEGO^®^-based therapy, pediatric epilepsy, neurological disease, cognitive habilitation, robotic programming

## Abstract

In the pediatric population, epilepsy is one of the most common neurological disorders that often results in cognitive dysfunction. It affects patients’ life quality by limiting academic performance and self-esteem and increasing social rejection. There are several interventions for the neurohabilitation of cognitive impairment, including LEGO^®^-based therapy (LEGO^®^ B-T), which promotes neuronal connectivity and cortical plasticity through the use of assembly sets and robotic programming. Therefore, the aim of this study was to analyze the effect of LEGO^®^ B-T on cognitive processes in pediatric patients with epilepsy. Eligible patients were identified; in the treatment group, an initial evaluation was performed with the NEUROPSI and BANFE-2 neuropsychological tests. Then, the interventions were performed once a week, and a final test was performed. In the control group, after the initial evaluation, the final evaluation was performed. An overall improvement was observed in the LEGO^®^ B-T patients, with a significant increase in BANFE-2 scores in the orbitomedial, anterior prefrontal, and dorsolateral areas. In addition, in the gain score analysis, the orbitomedial and memory scores were significantly different from the control group. LEGO^®^ B-T neurohabilitation is a remarkable option for epilepsy patients, who are motivated when they observe improvements.

## 1. Introduction

Epilepsy is a neurological disorder that affects many people, and a recent study reported an incidence of 61.4 per 100,000 person-years and a prevalence of 7.60 per 1000 people [1]. Among the pediatric population, one of the most important neurological diseases has an incidence of 41–187 per 100,000 people, which decreases to adult levels by the end of the first decade [2]. According to the International League Against Epilepsy (ILAE) in 2017, the operational epilepsy classification is focal, generalized, combined (generalized and focal), or unknown [3]. Focal seizures are the predominant seizure type in both children and adults [1].

The main neurophysiological feature of this pathology is the imbalance between the excitation and inhibition systems in the brain, with a predominance of excitotoxicity, hyperexcitability, and hypersynchrony of the neuronal networks in the brain [4,5]. In addition, epilepsy often leads to cognitive dysfunction in approximately 70–80% of patients, but treatment options are limited [6,7]. Previous studies have shown that multiple factors affect cognitive function in epileptic patients, including epilepsy itself, epilepsy treatment, uncontrolled seizures, psychosocial consequences of epilepsy such as stigma and marginalization, structural brain deficits, brain lesions, and individual reserve capacity [8,9]. This condition leads to behavioral or neuropsychiatric problems and impaired psychosocial functioning, resulting in learning disabilities, particularly in terms of memory, attention, mental processing speed, and other executive functions such as flexibility, emotional control, initiation, planning, organization, monitoring, and self-control, which involve subcortical and cortical brain structures [10,11]. Notably, in a sample of patients with newly diagnosed, untreated epilepsy, nearly 75% had deficits in attention, executive functions, and memory [12,13]. In another study, 39.5% of patients with epilepsy had visual memory deficits, 23.7% had attention and executive function deficits, and 15.8% had visual memory and language deficits [14].

Moreover, the origin of epilepsy or seizures may be inflammation, hypoxia, and brain immaturity in preterm infants, and pharmacological methods to prevent or treat brain injury and/or cognitive dysfunction at this stage have been proposed in preclinical models or clinical trials. The use of antenatal steroids and magnesium sulfate stabilizes the fetal neurovasculature and induces neuroprotection, reducing the risk of cerebral palsy [15,16,17,18]. In postnatal treatment, the use of pluripotent cells promotes angiogenesis, neurogenesis, synaptogenesis, and neurite outgrowth, improving behavioral deficits [19,20,21,22]; the use of erythropoietin and melatonin reduces inflammation and oxidative stress and promotes neurogenesis and angiogenesis, reducing learning deficits after brain injury [23,24,25,26,27]; caffeine citrate prevents apnea of prematurity and reduces cerebral palsy and cognitive delay [28,29,30].

Other non-pharmacological methods of neuroprotection have been used because of their impact on both preventing acquired brain injury and promoting healthy brain maturation. Kangaroo care or skin-to-skin care has been shown to reduce premature mortality and has been associated with improved biomarkers of resilience and stress reduction in infants, as well as influencing neurodevelopment [31,32,33]. Maternal breastfeeding has been associated with improved brain health outcomes and neuroprotective care [34,35]. Other infant neuroplasticity interventions that improve long-term cognitive or motor outcomes include the family-centered care model [36], massage therapy [37,38], music therapy and reading programs [39], improving physiological sleep-wake cycles [40], and positive social interactions [41]. Finally, a systematic review of recent clinical trials identified the efficacy of the acetylcholinesterase inhibitors donepezil and galantamine, the noncompetitive N-metil-D-aspartato antagonist memantine, and the stimulant methylphenidate, as well as noninvasive, nonpharmacological transcranial magnetic stimulation and transcranial direct current stimulation, as current strategies for treating cognitive dysfunction and improving language, memory, and attention [42].

In the context of cognitive habilitation, there are several interventions for cognitive impairment neurohabilitation, such as cognitive training, cognitive stimulation, and cognitive rehabilitation; some are computer-based, while others are noncomputer-based or mixed [43]. One of these neurohabilitation methods is LEGO^®^-based therapy (LEGO^®^ B-T), which promotes neuronal connectivity and cortical plasticity through the use of assembly sets and robotic programming, with the goal of improving neurocognitive processes [44]. This method involves multiple assembly and gearing steps aimed at promoting the cognitive work of various processes, such as attention, memory and executive function [45]. These robotic models begin with simple assemblies that increase in complexity as the LEGO^®^ B-T progresses [46,47]. Several studies have supported these therapeutic benefits, such as an increase in social skills in pediatric autism patients and a significant increase in executive and frontal functions in congenital heart disease patients [45]. The aim of the present study was to analyze the effect of LEGO^®^ B-T on attention and executive functions, memory, and attention and memory, as well as the orbitomedial cortex (OMC), anterior prefrontal cortex (APC), dorsolateral cortex (DLC), and total executive functions in pediatric patients with epilepsy using the neuropsychological batteries NEUROPSI and BANFE-2.

## 2. Materials and Methods

### 2.1. Study Design, Settings, and Ethical Considerations

This was a quasi-experimental study carried out in pediatric epilepsy patients at the Neurohabilitation and Behavior Unit of the Instituto Nacional de Pediatría (INP). The sample was selected intentionally, at convenience, and with voluntary participation. Patient recruitment was carried out from July 2022 to October 2023. All patients and parents were adequately informed about the study objectives, methods, likely benefits, foreseeable risks, and inconveniences. Written informed consent was obtained from each patient.

The protocol was approved by the Institutional Research and Ethics Committee of the INP (registration number 2022/45). The guidelines of human experimentation were followed in accordance with the ethical standards of the Institutional Research Committee and the code of ethics of the World Medical Association Declaration of Helsinki.

### 2.2. Patients

Patients with a diagnosis of epilepsy in the Neurology Service with available electronic medical records were enrolled. To recruit patients, the LEGO^®^ B-T intervention was first explained to the parents of the children before they applied to any test, and they were invited to participate. The parents and their children were free to decide whether to participate. If they agreed, they were in the LEGO^®^ B-T group, and if they did not want to, they were only given the neuropsychological tests once and six months later. So, although the selection of patients was not completely random, the study was blinded. Initially, 45 patients agreed to participate in the study, but only 22 patients completed it—12 in the LEGO^®^ B-T group and 10 in the control (CTRL) group—with ages ranging around 10.33 ± 2.96 and 11.9 ± 3.17 years for the LEGO^®^ B-T and CTRL groups, respectively. Each patient was considered according to the inclusion, exclusion, and elimination criteria. Inclusion criteria were as follows: patients with cognitive deficits (in at least one of the evaluated areas) associated with epilepsy; patients aged between 6 and 18 years; patients who agreed to participate in the study and who provided assent and informed consent; and patients whose parents or guardians agreed and signed the informed consent form for children under 12 years of age. Exclusion criteria were patients with cognitive deficits associated with epilepsy who suffered from any psychiatric disorder or any relevant psychological disorder. Elimination criteria were as follows: patients who (or their parents/guardians) decided to withdraw from the study at any time, after signing the informed consent form; patients who did not attend their follow-up visit within a period longer than 4 weeks after the date recorded; and patients who did not complete the neuropsychological habilitation treatment. The evaluations and interventions were conducted by LEGO^®^ Education-trained neuropsychologists.

### 2.3. Instruments

#### 2.3.1. Neuropsychological Attention and Memory Battery (NEUROPSI)

The NEUROPSI is a brief, objective, and reliable neuropsychological assessment instrument that evaluates cognitive processes in neurological patients. It was developed and validated in Mexico and allows cognitive disorders detection in a wide age range of the population. It consists of a series of tests designed to assess global cognitive functioning with 3 scales, 4 subscales and 32 exercises [48]. Attentional processes are evaluated with the subscales of selective and sustained attention and attentional control, and memory types and stages with subscales of working memory, short- and long-term memory, and verbal and visuospatial information. Finally, the battery provides totals for attention and executive functions, memory, and memory and attention. The scheme consists of simple and short items that are used depending on the total number of subscales. The natural points of the subtests are converted into normalized scores with a mean of 10 and a standard deviation of 3, with high reliability and a Cronbach’s alpha greater than 0.80 in the Mexican population from 6 years of age. Interpretation of the total score classifies an individual’s performance as high normal (116 and above), normal (85–115), moderate impaired (70–84), or severely impaired (less than 69) [48].

#### 2.3.2. Neuropsychological Battery of Executive Functions (BANFE-2)

The BANFE is a comprehensive, accurate neuropsychological evaluation that is suitable for children and has been validated in the Mexican population. It is an instrument that combines a significant number of tests with high reliability and validity for cognitive processes evaluation that depend mainly on the prefrontal cortex. It allows us to obtain a global index of performance in the battery and an index of the functioning of the three prefrontal areas evaluated: the OMC, the APC, and the DLC; it indicates the abilities and inabilities of children in each of these cognitive areas [49]. The OMC is associated with motor control, inhibitory control, and risk selection; the APC evaluates abstract meaning, metamemory and metacognitive control; the DLC assesses working memory, consecutive and inverse operations, planning, visuospatial working memory, visuospatial planning, abstract reasoning productivity, cognitive flexibility, verbal fluency, and sequential planning [49]. The instrument is valid in the Mexican population from 6 years of age. The natural points were converted into normalized scores and had a mean of 100 and a standard deviation of 15. The interpretation of the total score and each of the areas allows us to classify an individual’s performance as follows: high normal (116 and above), normal (85–115), moderate mild impairment (70–84), or severe impairment (less than 69), with high reliability and a Cronbach’s alpha greater than 0.80 [49].

#### 2.3.3. Evaluation of the Execution of LEGO^®^-Based Therapy

The intervention was evaluated with a Likert-type scale that was constructed according to the neuropsychological functions (which must be consolidated from the age of 6) and the batteries variables [49]. The scores were classified as follows: 0 = “does not execute”, 1 = “difficult to execute”, 2 = “executes”, and 3 = “easily executes”, with rating scores ranging from 0.80 to 0.90, indicating acceptable validity and agreement coefficients [45,50]. The percentage of performance of each patient in each session was then obtained by adding the different scores for each exercise, multiplying by 100, and dividing by the maximum total score.

### 2.4. General Procedure

After the relevant clinical and demographic data registration from the patients recruited in the study, an initial evaluation of the cognitive deficit degree was performed using the neuropsychological tests NEUROPSI (120 min long) and BANFE-2 (60 min long). The initial evaluations were performed one day per week (60 min long) for 3 weeks. Then, in the CTRL group, a second appointment was scheduled after 6 months for the final evaluation, while in the LEGO^®^ B-T group, the children performed between 7 and 18 interventions (depending on their progress in each session) of 60 min once a week until their postintervention evaluation.

### 2.5. Sessions of Intervention through LEGO^®^-Based Therapy

Briefly, Table 1 shows the descriptions with LEGO^®^ B-T sessions. The interventions were conducted individually in the LEGO^®^ B-T group according to the study reported by Espinosa-Garamendi et al., (2022) [45], using LEGO^®^ Education sets and the LEGO^®^ scale to assess patient execution [45,50].

### 2.6. Statistical Analysis

The observed scores for patients in the experimental setting were plotted and colored by etiology and type of epilepsy. The dependent variables examined were attention and executive function; memory and attention and memory; OMC, APC, DLC, and total executive function. Analysis of covariance (ANCOVA) [51,52] was used to test the hypothesis that scores were significantly different between the CTRL and treatment groups, followed by Bonferroni post hoc correction. In addition, the gain score was examined using the Wilcoxon rank sum test [53]. Statistical analysis was performed using R version 3.4.1 and RStudio version 0.99.902 software. Plots were generated using the R package version 4.1.1 [54]. A *p* value < 0.05 was considered statistically significance.

## 3. Results

### 3.1. Descriptive Analysis of the Population

The descriptive characteristics of pediatric patients with epilepsy are shown in Table 2. A total of 22 patients were analyzed, 10 in the CTRL group and 12 in the LEGO^®^ B-T group. The age range was between 6 and 15 years in the CTRL group and between 7 and 15 years in the LEGO^®^ B-T group, with a median age of 12 years in the CTRL group (interquartile range, IQR 10–15) and 9.5 years in the LEGO^®^ B-T group (IQR 8–12.5). In addition, we classified the etiology and type of epilepsy according to Scheffer 2017 [3], and it was observed that in both groups, the CTRL and LEGO^®^ B-T groups had structural, genetic, and unknown etiologies. However, there were more patients with structural etiology in the CTRL group (n = 8) and more patients with unknown etiology in the LEGO^®^ B-T group (n = 6). Also, most patients in the CTRL and LEGO^®^ B-T groups had focal (n = 7 and n = 5, respectively) or combined epilepsies (n = 3 and n = 5, respectively) with or without paroxysmal activity in their electroencephalograms (EEGs).

Table 3 shows the treatments prescribed for each patient. Notably, in both the CTRL and LEGO^®^ B-T groups, levetiracetam (LEV) was the most commonly used drug (with seven children in each group), followed by valproic acid (VPA; n = 4; n = 2, respectively) and oxcarbamazepine (OXCBZ; n = 1; n = 5, respectively). The least commonly used drugs were lamotrigine (LTG; n = 2; n = 1, respectively) and topiramate (TPM; n = 1 in the CTRL group). Treatment was recently discontinued for one patient.

### 3.2. Cognitive Evaluations of Pediatric Patients with Epilepsy

The normalized scores of the cognitive evaluations of pediatric patients with epilepsy are shown in Table 4. The evaluations were performed with the Neuropsychological Attention and Memory (NEUROPSI) Test to evaluate attention and executive functions, memory, and attention and memory, and with the Neuropsychological Battery of Executive Functions (BANFE-2) to evaluate OMC, APC, DLC, and total executive functions.

According to the NEUROPSI battery, for attention and executive function, the scores increased in six CTRL and nine LEGO^®^ B-T patients (with changes in clinical diagnosis in four and five patients, respectively); scores were the same before and after evaluation in three CTRL and two LEGO^®^ B-T patients, and scores decreased in one CTRL and one LEGO^®^ B-T patient. For memory, however, scores increased in three CTRL and ten LEGO^®^ B-T patients (with changes in clinical diagnosis in two CTRL and seven LEGO^®^ B-T patients), and the scores were the same before and after the evaluation in seven CTRL and two LEGO^®^ B-T patients. For attention and memory, scores increased in four CTRL and ten LEGO^®^ B-T patients (with a change in clinical diagnosis in three CTRL and five LEGO^®^ B-T patients); they were the same before and after evaluation in five CTRL and two LEGO^®^ B-T patients, and decreased in one CTRL patient.

On the other hand, according to the BANFE-2 test, in OMC, the scores increased in six CTRL and eleven LEGO^®^ B-T patients (with changes in clinical diagnosis in four CTRL and nine LEGO^®^ B-T patients); scores were the same before and after evaluation in three CTRL and one LEGO^®^ B-T patient, and scores decreased in one CTRL patient. For APC, the scores increased in five CTRL and nine LEGO^®^ B-T patients (with changes in clinical diagnosis in five CTRL and eight LEGO^®^ B-T patients), and the scores were the same before and after the evaluation in two LEGO^®^ B-T patients. The score decreased in five CTRL patients and one LEGO^®^ B-T patient. For DLC, scores increased in five CTRL and twelve LEGO^®^ B-T patients (with a change in clinical diagnosis in four CTRL and five LEGO^®^ B-T patients), were unchanged in two CTRL patients, and decreased in two CTRL patients. Finally, for total executive function, the normalized scores increased in five CTRL and eleven LEGO^®^ B-T patients (with clinical diagnosis changes in four CTRL and six LEGO^®^ B-T patients); in four CTRL and one LEGO^®^ B-T patient, the scores were the same before and after the evaluation.

In addition, Figure 1 shows the scatterplots between the pretest and the posttest by group for each variable of the cognitive evaluations of pediatric patients with epilepsy. The graphs present the data points colored by group, the fitted regression lines (solid lines) for both scores by group, and their 95% confidence intervals (shaded areas). In general, the scatterplots show a linear relationship between the pretest and posttest scores, with increasing trends in both groups. Note that the confidence intervals in the CTRL group are wider than those in the LEGO^®^ B-T group because the CTRL group has fewer subjects and the data are more scattered.

Moreover, several of the characteristics show similar results; in particular, the graphs for the variables attention and executive function, attention and memory, and total executive functions show both estimated lines such that the LEGO^®^ B-T ones are greater than the CTRL ones; however, their confidence bands intersect and overlap almost completely in the areas where there are scores, indicating that there is no evidence that the scores in the two groups are different. For the variable memory, the graph shows that the estimated line and the confidence interval intersect in some areas but not completely in the whole area where there are points; therefore, an exhaustive analysis is necessary to prove or reject the significant difference between the two groups (see below). For the variables OMC, APC, and DLC, the graphs show that the estimated lines and the confidence intervals are more separated, which means that the posttest scores are different between the groups, given their pretest scores, and that the results for the LEGO^®^ B-T group are greater.

### 3.3. Execution of LEGO^®^-Based Therapy for Pediatric Patients with Epilepsy

The mean percentages of LEGO^®^ B-T group execution by session is shown in Figure 2, and each line is colored according to etiology (top) or epilepsy type (bottom). Notably, there was a significant scatter in the data for three main reasons. The first is that, in some cases, there were children with greater cognitive deficits who found it more difficult to perform the exercises in some sessions. The second was that the children experienced a change in medication or dosage and became lethargic or drowsy. The third reason was that the children missed two sessions in a row, and the previous progress decreased; however, it is possible to observe an improvement in the execution of the LEGO^®^ B-T, with the exception of week 15.

### 3.4. Effect of LEGO^®^ B-T in Pediatric Patients with Epilepsy

To evaluate the overall improvement in attention and executive function, memory, and attention and memory in LEGO^®^ B-T in pediatric patients with epilepsy, we used the NEUROPSI test. The results showed a tendency for the LEGO^®^ B-T group to increase their scores on the posttest compared to the pretest in all the processes evaluated, especially in memory (Figure 3, left). In addition, an increase in the scores of the experimental group compared to those of the CTRL group was also observed (Figure 3, right). However, no statistically significant differences were found.

In addition, Figure 4 summarizes the results of the evaluation of the effect of LEGO^®^ B-T on the function of OMC, APC, and DLC with the BANFE-2 battery. Again, comparing the pretest versus posttest normalized scores of each group, the results show an increase in the LEGO^®^ B-T group in all evaluated functions (Figure 4, left) and an increase in the scores of the experimental group compared to the CTRL group (Figure 4, right), with statistically significant differences in the OMC, APC, and DLC areas.

### 3.5. Analysis of the Gain Score

Another procedure to analyze the effects of LEGO^®^ B-T in pediatric patients with epilepsy (evaluated with the NEUROPSI and BANFE-2 tests) is to compare gain scores, which evaluate the differences between the pretest and posttest for each variable, i.e., gain score = posttest score-pretest score [51]. The results showed that the CTRL and LEGO^®^ B-T groups were significantly different for memory and OMC (Figure 5). For memory, the median of the CTRL group was 0, and the first and third quartiles were 0 and 10; however, for treatment, the median was 18.5, and the first and third quartiles were 7 and 33.5, respectively. For OMC, the median of the CTRL group was 4, and the first and third quartiles were 0 and 28; however, for the treatment group, the median was 30.5, and the first and third quartiles were 21.5 and 38, respectively. The boxplots show a significant difference between the CTRL and treatment groups, with the posttest scores in the treatment group being greater than those in the pretest scores in these two areas.

## 4. Discussion

Regarding the characteristics of the epileptic population studied, we observed a greater number of patients with structural etiology, which is consistent with the epidemiological results of other clinical studies [1,55]. In addition, the present study showed that both groups of patients had mild-moderate to severe alterations in most cognitive functions at baseline, such as attentional memory and executive functions. These results are also consistent with those reported in other studies of learning and intelligence quotient impairments in pediatric patients with epilepsy [10,11]. It should be noted that several LEGO^®^ B-T patients, were observed to transition to a normal or high normal diagnosis.

The results showed an upward trend in the interventions; however, this trend was consistently observed in epileptic patients of unknown etiology and generalized epilepsy. For other etiologies, the same ascending pattern was observed, but especially in the intermediate sessions, the degree of evocation decreased. According to the above, it is necessary to consider several sessions of neuronal cognitive work to adequately observe the complete evolution of the patient; in addition, the level of cognitive effort reversal and the evolution of the epileptic patient during the intervention must be evaluated, as we did. In addition, variables such as patient rest and fasting, the presence of headaches, and recurrence should be considered since habilitation with LEGO^®^ sets is sensory and cognitively stimulating [56].

In addition, a significant increase in the numerical ranges of the BANFE-2 (OMC, APC, and DLC) and NEUROPSI (memory) batteries was observed in the results obtained in the experimental group compared to those obtained in the CTRL group. This finding is consistent with previous results obtained by our group in relation to the habilitation of basic frontal functions and executive functions in pediatric patients with congenital heart disease. In these patients, a variation in cognitive deficit was also observed, as well as the number of therapies required to improve functions due to the etiological subclassification of the sample [45,50].

Cognitive habilitation produced a significant increase in the subprocesses associated with the APC (thought abstraction and metamemory) and in the OMC (inhibitory control, risk selection, and maintenance of positive responses). The results obtained suggest that the LEGO^®^ B-T enhances the development of frontal functions and, thus, the management of behavior. However, it should be mentioned that for future research, a greater number of exercises related to the DLC should be added, especially in arithmetic and reading comprehension. Regarding the attention and orientation process, a nonsignificant increase in numerical range scores was obtained. Taken together, these results suggest that the intervention improves this circuit but targets the inhibitory process; however, future research should work on the vigilance and sustained attention network [48].

We should mention that another novelty of the LEGO^®^ B-T is that marginal means with a tendency to significance were obtained in the total memory process, which indicates that by adding exercises related to evocation and retrieval, working memory as well as short- and medium-term memory were stimulated and enabled. These exercises consisted of working with sets to remember and recognize figures as well as assembling them without templates, which also generated fun and excitement in the patients since they were challenging their memory themselves.

An interesting result was observed in the CTRL group at the end of the second evaluation. There were no significant increases in the diagnostic ranges, but there were significant increases in the numerical ranges, indicating that there was a clinical change in the neuropsychological batteries. These results suggest that although cognitive habilitation did not occur in this study group, the cerebral cortex generates cortical plasticity with specialization of neurons during normal development in pediatric patients [57,58]. Therefore, it is important to stimulate cognitive functions beginning in childhood that can be consolidated during neurodevelopment. The positive effects observed with LEGO^®^ B-T were not only observed in cognitive functions but also in the development and acquisition of social skills, sensorimotor functions, and emotions, as shown in other studies. In studies with long-term LEGO^®^ B-T with humanoid robots in children with autism spectrum disorder, improvements in attention were observed, suggesting that LEGO^®^ B-T should be used as a mediator in social skills training for autistic children as LEGO^®^ B-T improves collaborative behaviors such as initiating interaction, responding, and playing together [59,60,61]. In children with cerebral palsy, the use of Chinese puppets with LEGO^®^ robots improved range of motion, finger tapping test scores, and finger pressing speed after puppet therapy [62]. Other studies have shown that LEGO^®^ bricks improve resilience in gifted children [63] and reduce stress and increase socialization in university students [64]. On the other hand, in elderly people, it has been observed that the use of an interactive format of the LEGO^®^ robot with a tablet has been observed to produce physical and cognitive improvements related to memory and mathematical problem-solving in elderly people [65,66]. In addition, the application of sensory stimulation to poststroke individuals using haptic object recognition with LEGO^®^ bricks improved tactile coactivation and was shown to be a useful therapeutic intervention for sensory deficits following stroke [67]. The use of LEGO^®^ B-T for sensorimotor training in Parkinson’s idiopathic patients also significantly improved tactile information through haptic performance [68]. In adults who had experienced domestic childhood violence, the use of LEGO^®^ SERIOUS PLAY^®^ showed that the subjects were able to embed inner strengths and develop strengths in a relationship as evidenced by a consistent increasing trend on their Critical Positivity Ratio Self-Test Scale, a positive change trend on the Attachment-Related Avoidance Scale post-test, and an increasing trend in their scores on the Attachment-Related Anxiety Scale [69]. This same intervention has also been used with economically disadvantaged students, with decreased anxiety and depression and increased well-being [70].

Regarding the effect of pharmacotherapy on cognitive performance in children with epilepsy recruited for this study, it has been observed that the use of antiepileptic drugs can have side effects on the cognitive performance of these children, as their use affects the development of the central nervous system. The most commonly used antiepileptic drugs have been linked to cognitive deficits and cause learning problems. Phenobarbital has been shown to alter attention and memory processing, phenytoin has been shown to affect the speed of cognitive processing, while the effects of OXCBZ and carbamazepine have not been clarified [71,72,73]. Topiramate has been shown to cause deficits in attention, memory, and language [74,75,76]. In the case of LEV, vigabatrin and ethosuximide appear to have no side effects, although there is no conclusive evidence; however, LEV is associated with psychobehavioral side effects, including aggressiveness and irritability [76]. In the case of VPA and LTG, improvements in attention have been reported, although other studies have found LTG to be associated with insomnia [73,76]. Studies on gabapentin, tiagabine, zonisamide, and rufinamide in pediatric patients have not yet been reported [77,78]. Based on the above, we can assume that the beneficial effects on cognitive functions in both groups of patients are mainly due to LEGO therapy, considering that the effect of LEV (administered to at least 70% of the enrolled patients) on cognitive functions has not yielded conclusive results.

Finally, as reported in other studies, LEGO^®^ B-T has proven to be a good tool for cognitive and social neurohabilitation in patients with congenital heart disease, patients with autism, and elderly people [44,45,47,65]. In clinical practice, the perspectives are to promote the development of communication (linguistic, operational, social, strategic) and emotional well-being, to develop social skills such as turn taking, joint attention, sharing, joint problem solving, and listening in the epileptic children, and to reduce the costs by reducing the time required for resources personalization. Furthermore, it is necessary to study other pathologies and adjust the sessions for each patient. In addition, it is necessary that pediatric patients can have fun while being habilitated with new therapeutic tools that support neurodevelopment to achieve better academic opportunities in the future.

## 5. Limitation of the Study

It is important to point out the limitations of this study due to the small sample size. In general, it is necessary to have sufficiently large sample sizes to guarantee that the results are statistically significant. However, under the ANCOVA test, we consider that the sample size used in this paper may be within the appropriate minimum to guarantee the significant differences that we detected since there are some articles that support the idea that using ANCOVA can considerably reduce the sample size required to detect significant differences. For example, Borm et al., (2007) [79] proposed a formula to compute the sample size for ANCOVA, and through it, we concluded that with between 8 and 20 patients, we could find significant differences in OMS, APC, DLC areas, and memory. In addition, Shan and Ma (2014) [80] compare the approach of [79] with an exact approach, using the noncentral F distribution, for analysis of covariance with one covariate, with similar results. Another more recent work on sample size was carried out by Bujang et al. (2017) [81] for linear regression models and ANCOVAs. They estimated the minimum sample size required in these two tests when R-squared was used as the effect size. Although they suggest that a sample size of 300 is necessary for clinical trials, they generate tables for the different R square values for control and treatment groups and, for example, for the OMC with R square of the CTRL and LEGO^®^ B-T given by R-squared_CTRL = 0.78 and R-squared_LEGO = 0.57, with a single variable for the group, a sample size of 10 would be enough to show significant differences. Moreover, there are studies that have used interventions with LEGO therapy and have also recruited few patients (fewer than 10) to demonstrate the effect of the therapy on neuropathologies [59,60,63,65,66,69].

## 6. Conclusions

In the present study, we observed that cognitive habilitation of pediatric epilepsy patients using LEGO^®^ B-T resulted in significant changes in the function of OMC, APF, and DL areas and in memory. By increasing the functions associated with these regions, the children were able to improve their cognitive abilities, promoting learning capacity in academic and functional areas of daily life. Neurohabilitation with LEGO^®^ B-T is a remarkable therapeutic method that motivates patients and their parents when they see improvements.

## Figures and Tables

**Figure 1 brainsci-14-00702-f001:**
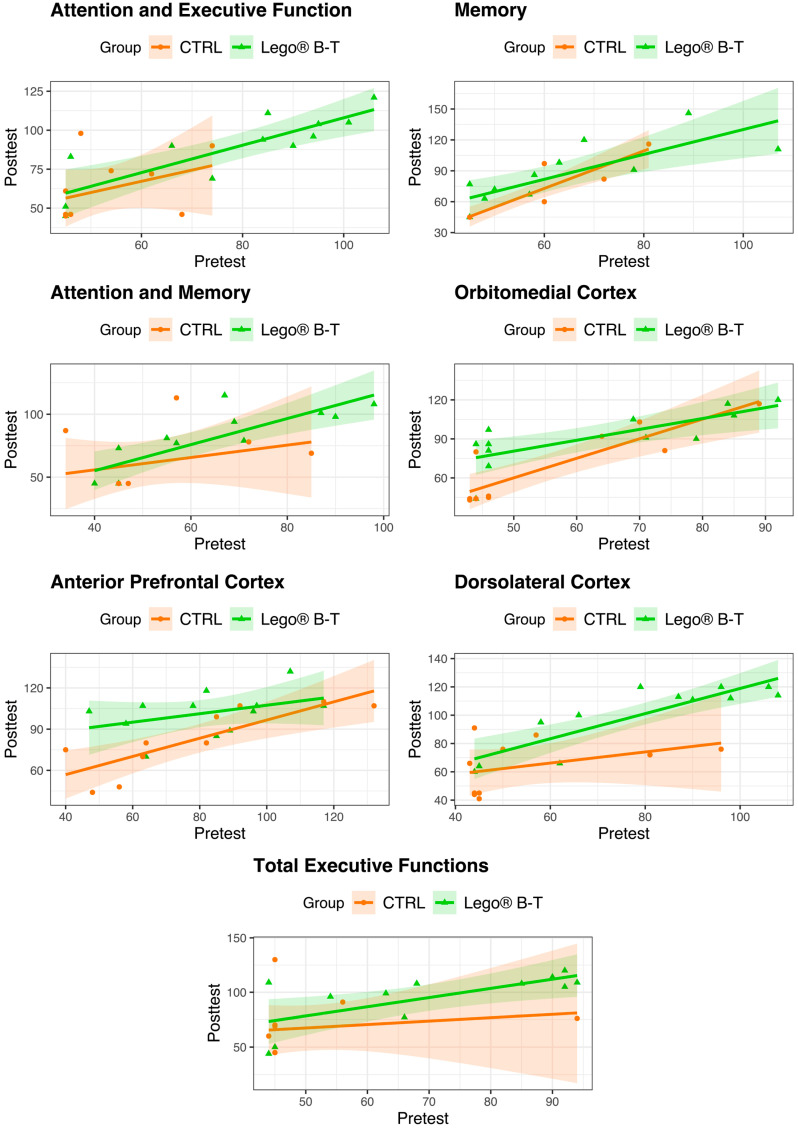
Scatter plots between the pretest and posttest scores by group, showing the fitted regression lines (solid lines) and their 95% confidence intervals (shaded areas) by group. CTRL = control; LEGO^®^ B-T = LEGO^®^-based therapy. CTRL n = 10; LEGO^®^ B-T n = 12.

**Figure 2 brainsci-14-00702-f002:**
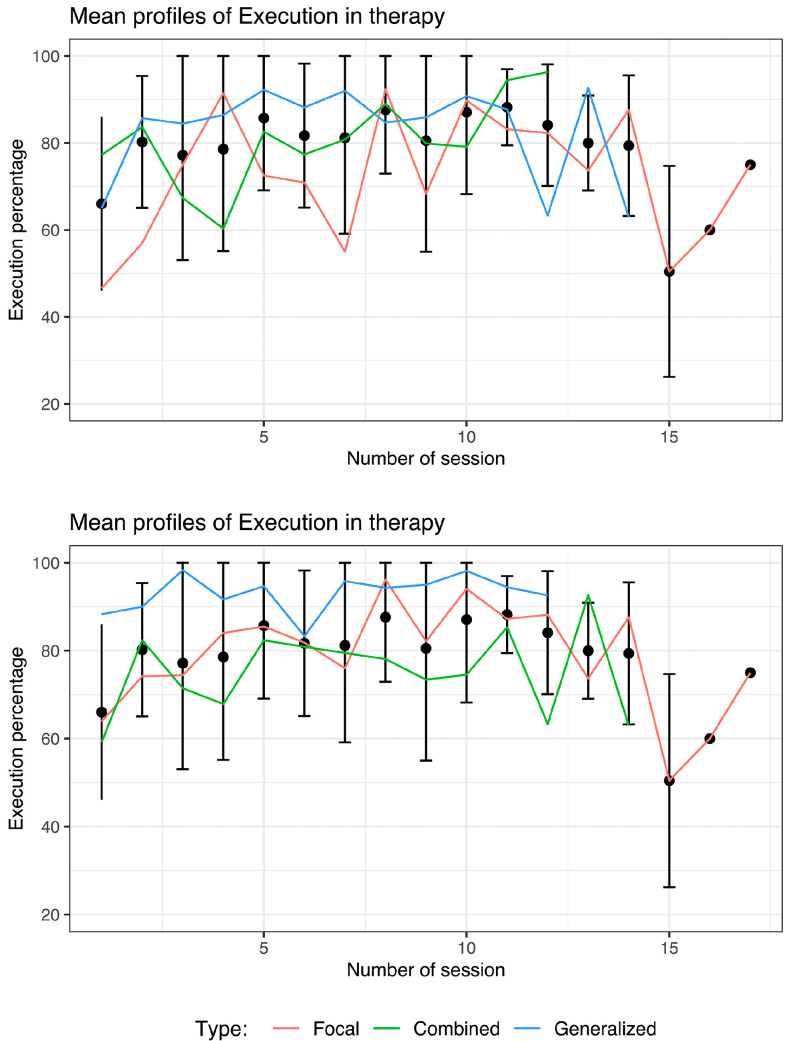
Evaluation of LEGO^®^ based therapy execution by session. Each line is colored according to etiology (**top**) or epilepsy type (**bottom**).

**Figure 3 brainsci-14-00702-f003:**
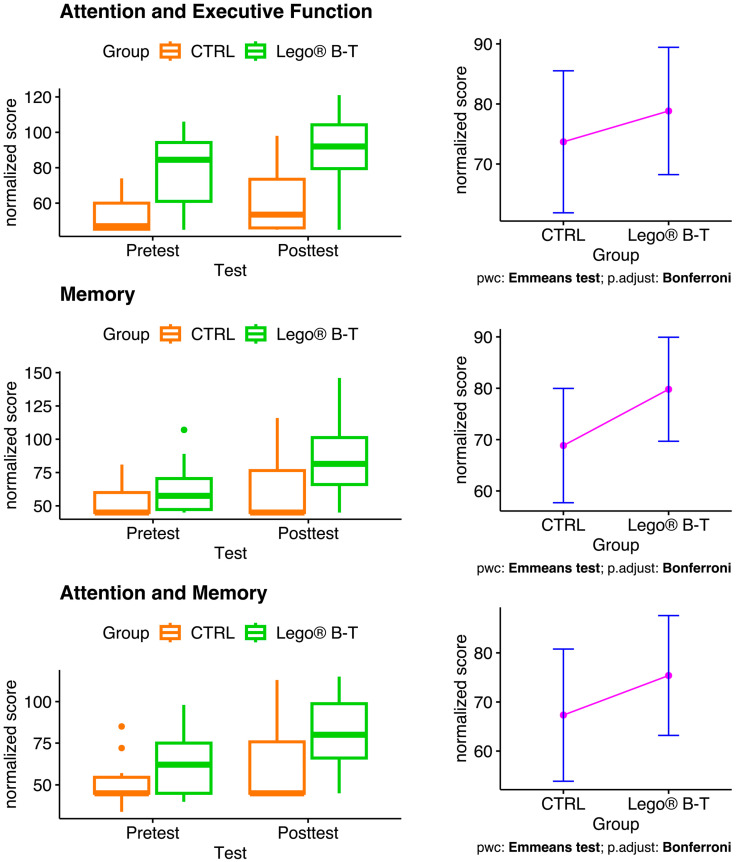
Effect of LEGO^®^ based-therapy in pediatric patients with epilepsy as evaluated by NEUROPSI test. The box plots on the left show the changes in scores in the control (CTRL) and treatment groups. The plots on the right show the pairwise comparisons (pwc), the estimated marginal means of the posttest scores by group under the mean of the pretest. Note an increase in the treatment group in all areas evaluated. CTRL = control; LEGO^®^ B-T = LEGO^®^-based therapy. CTRL n = 10; LEGO^®^ B-T n = 12.

**Figure 4 brainsci-14-00702-f004:**
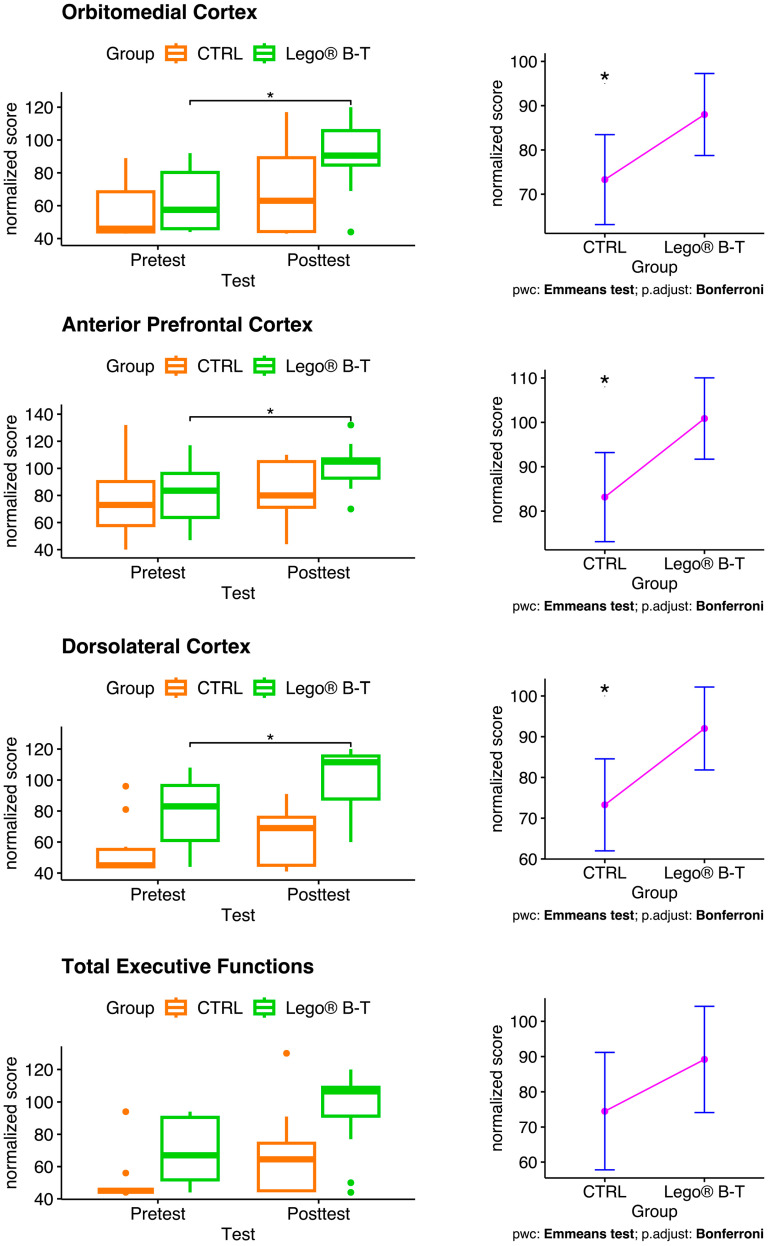
Effect of LEGO^®^ based-therapy in pediatric patients with epilepsy as evaluated by the BANFE-2 test. Box plots on the left show changes in scores in the control (CTRL) and treatment groups. There was a significant increase in scores after LEGO^®^ B-T in the OMC, APC, and DLC. ANCOVA test; * *p* ≤ 0.050. The graphs on the right show the pairwise comparisons (pwc), the estimated marginal means of the posttest scores by group under the mean of the pretest. ANCOVA followed by the Bonferroni post hoc correction with correction for multiple testing. CTRL = control; LEGO^®^ B-T = LEGO^®^-based therapy. CTRL n = 10; LEGO^®^ B-T n = 12.

**Figure 5 brainsci-14-00702-f005:**
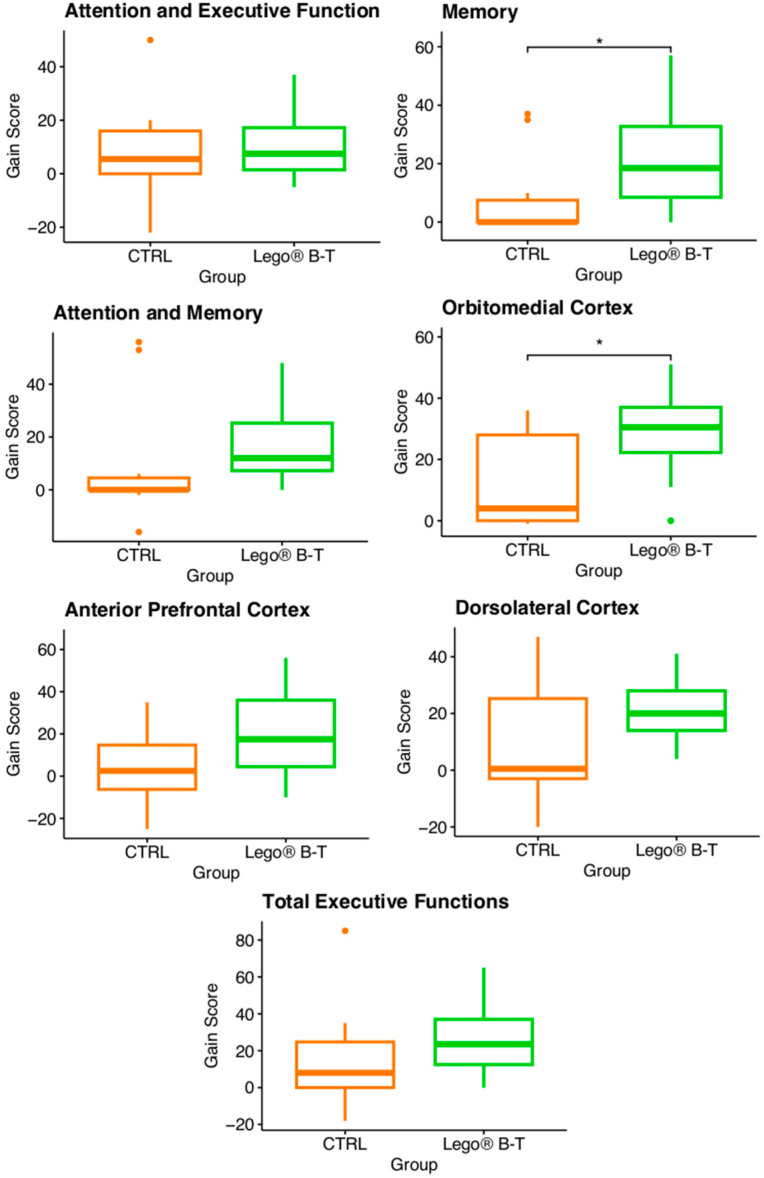
Effect of LEGO^®^-based therapy on the gain score of pediatric patients with epilepsy, as evaluated by the NEUROPSI and BANFE-2 tests. The box plot shows the gain scores for the variables evaluated by group. There was a significant difference between the CTRL and treatment groups for memory and OMC. Wilcoxon test; * *p* ≤ 0.05. CTRL= control; LEGO^®^ B-T = LEGO^®^-based therapy. CTRL n = 10; LEGO^®^ B-T n = 12.

**Table 1 brainsci-14-00702-t001:** Intervention of cognitive functions with LEGO^®^-based therapy.

Sessions	Objectives	Tasks	Sets
(1) Free game with programming	Initial interaction, familiarizing the patient with the material and conducting the therapeutic interaction	Identify colored blocks and free assembly Initial robot assembly and programming In severe cases, start assembling animal sets Simple machine assembly Programming of robotic challenges: forward and backward sequence at 10 s	▪ LEGO^®^ DUPLO^®^ bricks ▪ WeDo 2.0^®^ Set ▪ LEGO^®^ Simple machines Set ▪ Bingo LEGO^®^ Education Bingo Set
(2) and (3) Working and visuospatial memory, inhibitory control	Gradually stimulates areas of the cerebral cortex to improve selective attention, inhibitory control, and short-term memory	Begin with working memory training and assembling blocks of two or three pieces Stimulation using a visuospatial system Working memory template, blocks of the same color Assembly of turns and configuration of challenges Assembling and programming of robots with challenges Simple disassembly and reassembly of a machine or robot challenge without the assistance of a template or therapist	▪ LEGO^®^ DUPLO^®^ bricks ▪ WeDo 2.0^®^ Set ▪ LEGO^®^ Simple machines Set ▪ Assembled blocks template
(4) to (8) Working memory, inhibitory control, risk selection and planning	Stimulation of cerebral cortex areas to improve the effort investment process of related functions	Begin with working memory exercises and assembling blocks of three to four pieces Stimulate with visuospatial working memory template and color the blocks Classify blocks by color and label Assemble and program robots with a challenge	▪ LEGO^®^ DUPLO^®^ bricks ▪ WeDo 2.0^®^ Set ▪ Assembled blocks template
(9) to (16) Integration of follow-up and executive functions integration	Cognitive habilitation and developmental monitoring	Six-piece memory kit assembly Arming and assembling the robot Exercise of progressive and regressive mathematical order Complex task to solve with the robot Classification of animals set, concrete and abstract	▪ LEGO^®^ DUPLO^®^ bricks ▪ WeDo 2.0^®^ Set or SPIKETM Set ▪ LEGO^®^ Education Animals Set ▪ LEGO^®^ Education More to Math Set

**Table 2 brainsci-14-00702-t002:** Characteristics of pediatric patients with epilepsy.

Variable	Characteristic	CTRL Group	LEGO^®^ B-T Group
Sex ^1^	Male	4 (40%)	7 (58%)
	Female	6 (60%)	5 (42%)
Age ^2^	Years	12 (10–15)	9.5 (8–12.5)
Etiology ^1^	Structural	8 (80%)	4 (33%)
	Genetic	1 (10%)	2 (16%)
	Unknown	1 (10%)	6 (50%)
Epilepsy type ^1^	Focal	7 (70%)	5 (42%)
	Generalized	0 (0%)	2 (16%)
	Combined generalized and focal	3 (30%)	5 (42%)
EEG ^1^	With paroxysmal activity	5 (50%)	6 (50%)
	Without activity	4 (40%)	5 (42%)
	Not performed	1 (10%)	1 (8%)

^1^ Counts and percentages. ^2^ Median and interquartile range (1st quartile–3rd quartile). CTRL = control; LEGO^®^ B-T = LEGO^®^-based therapy; EEG = electroencephalogram.

**Table 3 brainsci-14-00702-t003:** Antiepileptic drugs and doses prescribed for pediatric patients with epilepsy.

CTRL			LEGO^®^ B-T Group		
Patient	AED	Dose (mg/kg/Day)	Patient	AED	Dose (mg/kg/Day)
1	LEV	37	1	VPA	19
2	LEV	27	2	OXCBZ	26
3	VPA	17	3	LEV	47
4	OXCBZ	25	4	LEV	27
5	LTG	4.2	5	LEV	36
6	LEV	30	6	LEV	40
7	LEV TPM	55 5.5	7	OXCBZ LTG	22 5.6
8	LEV VPA	20 46	8	LEV OXCBZ	50 30
9	VPA LEV	21.5 40	9	LEV OXCBZ	65 13
10	LEV	18	10	VPA	24
	LTG	3.5	11	Suspended	--
	VPA	31	12	OXCBZ LEV	30 42

AEDs = antiepileptic drugs; LEV = levetiracetam; LTG = lamotrigine; VPA = valproic acid; TPM = topiramate; OXCBZ = oxcarbamazepine.

**Table 4 brainsci-14-00702-t004:** NEUROPSI and BANFE-2 total normalized scores in control and LEGO^®^-based therapy of pediatric patients with epilepsy.

CTRL Patients with Epilepsy														
	NEUROPSI						BANFE-2							
Patient	Attention and Executive Functions		Memory		Attention and Memory		OMC		APC		DLC		Total Executive Functions	
	Pre	Post	Pre	Post	Pre	Post	Pre	Post	Pre	Post	Pre	Post	Pre	Post
1	62	**72**	60	**97**	34	**87**	89	**117**	64	**80**	44	**91**	45	**130**
2	45	61	45	45	45	45	64	**92**	117	**110**	43	66	45	69
3	54	**74**	60	60	85	**69**	44	**80**	63	**70**	50	**76**	45	70
4	68	46	45	45	47	45	44	44	132	**107**	81	72	44	60
5	45	46	45	45	45	45	43	43	56	48	45	41	45	45
6	48	**98**	81	**116**	57	**113**	70	**103**	85	99	57	**86**	56	**91**
7	45	45	45	45	45	45	46	46	82	80	44	44	45	45
8	46	46	45	45	45	45	43	44	48	44	45	45	45	45
9	46	46	45	45	45	45	46	45	40	**75**	44	45	45	45
10	74	**90**	72	82	72	78	74	81	92	107	96	**76**	94	**76**
**LEGO^®^ B-T patients with epilepsy**														
**NEUROPSI**							**BANFE-2**							
**Patient**	**Attention Executive Function**		**Memory**		**Attention and Memory**		**OMC**		**APC**		**DLC**		**Total Executive Functions**	
	**Pre**	**Post**	**Pre**	**Post**	**Pre**	**Post**	**Pre**	**Post**	**Pre**	**Post**	**Pre**	**Post**	**Pre**	**Post**
1	84	**94**	50	**72**	57	**77**	46	**97**	58	**94**	87	113	68	**108**
2	95	104	63	**98**	69	**94**	92	**120**	96	103	90	111	94	109
3	45	45	48	63	45	45	44	44	97	107	62	66	44	44
4	74	**69**	45	45	45	45	46	69	64	**70**	44	60	45	50
5	101	105	89	**146**	90	98	84	**117**	107	**132**	98	112	92	**120**
6	66	**90**	45	**77**	71	79	69	**105**	89	89	58	**95**	54	**96**
7	45	51	45	45	40	45	46	**86**	47	**103**	45	64	66	**77**
8	94	96	68	**120**	67	**115**	79	**90**	85	85	96	**120**	92	105
9	106	**121**	78	**91**	87	101	44	**86**	117	**107**	108	114	44	**109**
10	85	111	107	111	98	108	46	**81**	82	**118**	106	**120**	90	114
11	90	90	58	**86**	55	**81**	85	108	63	**107**	79	**120**	85	108
12	46	**83**	57	67	45	**73**	71	**91**	78	**107**	66	**100**	63	**99**

High normal = ≥116; normal = 85–115; mild moderate change = 70–84; severe change = ≤69. Numbers in bold indicate when the patient’s clinical diagnosis changed. CTRL = control; LEGO^®^ B-T = LEGO^®^-based therapy. OMC = orbitomedial cortex; APC = anterior prefrontal cortex; DLC = dorsolateral cortex. Changes in the diagnosis of cognitive deficits in post-test are in bold.

## Data Availability

The data presented in this study are available upon request from the corresponding author. The data are not publicly available due to privacy concerns.

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
