# Peer review of "Neurohabilitation of Cognitive Functions in Pediatric Epilepsy Patients through LEGO®-Based Therapy"

_brainsci, 2024, doi:10.3390/brainsci14070702_

Round 1

Reviewer 1 Report

Comments and Suggestions for Authors

This is a study of executive functions and various cognitive functions in pediatric patients with epilepsy. The aim of this study was to assess the effectiveness of LEGO®-based therapy on the executive and cognitive functions of children with various types of epilepsy.  The LEGO®-based therapy was introduced by this group of authors in 2012 and more recently in 2022 [doi:10.3390/healthcare10122348].  The details of LEGO®-based therapy are provided in Table 1.  I have a few critical comments that I would like to share.

1. In the present study, the authors used a neuropsychological test battery NEUROPSI, which is available in Spanish and BANFE-2, which might not be familiar to non-Mexican readers. BANFE-2 denotes "Batería de Funciones ejecutivas y lóbulos frontales" (Battery of Executive Functions and Frontal Lobes). Both test batteries were briefly described in the method section. In this section (lines 124-125), the abbreviations OMC, APC, and DLC made their enigmatic appearance. The authors are invited to disclose the first appearance of these abbreviations and explain the methodology used to define the functions of each cortical area.

2. The sample size was too low, and the inter-subject variability was very high. Therefore, the results of the study do not seem to be reliable. 

3. Line 268. "The effect of therapy in the CTRL and LEGO® B-T patients..." What kind of therapy did the control group receive? Figures 3 and 4 demonstrate difference between the pretest and the posttest results in the CTRL group.  As shown in Figure 4, the total executive function scores in the control group improved from the pretest to the posttest period (same to what is seen after LEGO®-based therapy). The similar dynamics of executive functions might be associated with the positive dynamics of epilepsy and the effectiveness of medical therapy in the control group. 

4. The initial results of the neuropsychological tests in the control group and the treated group were remarkably different (pretest values in Figures 3 and 4). The subjects in the treated group initially showed better attention and executive functions than CTRL group, as indicated by the results of the NEUROPSI test (Figure 3). They also demonstrated enhanced dorsolateral cortex functions in comparison to the CTRL group (Figure 4) and improved total executive functions, as measured by the BANFE-2 test (Figure 4). Therefore, the CTRL group in this study appears to be a distinct cohort.

4. The authors concluded (page 14) that "cognitive habilitation of pediatric epilepsy patients using LEGO® B-T caused significant changes in the OMC, APF and DL areas and in memory".  The aforementioned cortical areas were not examined in this study. The authors assessed the functions of these areas, but not the changes.

Minor.

The legends for Figures 3 and 4 are redundant in the text.

Figure 2 illustrates the marked variability in test executive scores between subjects. The top and bottom plots are the same, except for some differences in color. The authors are encouraged to consider a more effective illustration of the fact that "a patient with structural etiology and combined epilepsy and a patient with genetic etiology and focal epilepsy showed the lowest performance" (lines 282-284).   

Figure 1 and Table 3 present the same data. Figure 1 illustrates the underlying patterns and trends that appear to be quite similar in the CRTL and the treated group.

Author Response

Responses to reviewer 1.

This is a study of executive functions and various cognitive functions in pediatric patients with epilepsy. The aim of this study was to assess the effectiveness of LEGO®-based therapy on the executive and cognitive functions of children with various types of epilepsy.  The LEGO®-based therapy was introduced by this group of authors in 2012 and more recently in 2022 [doi:10.3390/healthcare10122348].  The details of LEGO®-based therapy are provided in Table 1.  I have a few critical comments that I would like to share.

  1. In the present study, the authors used a neuropsychological test battery NEUROPSI, which is available in Spanish and BANFE-2, which might not be familiar to non-Mexican readers. BANFE-2 denotes "Batería de Funciones ejecutivas y lóbulos frontales" (Battery of Executive Functions and Frontal Lobes). Both test batteries were briefly described in the method section.

R= Thank you for your comment. We added more information to the description of the NEUROPSI and BANFE-2 tests for better understanding on page 4.

  1. In this section (lines 124-125), the abbreviations OMC, APC, and DLC made their enigmatic appearance. The authors are invited to disclose the first appearance of these abbreviations and explain the methodology used to define the functions of each cortical area.

R= The abbreviations have been carefully reviewed and corrected. Regarding the methodology used to define the functions of each cortical area, during the development of neuropsychology, according to the functions associated with the cerebral cortex and the classification of areas, cognitive functions have been defined. For example, through neuroimaging studies, a subject is exposed to a cognitive task, and their performance is observed, as well as the brain activity underlying behavioral performance in the proposed task. Thus, external stimuli cause each cell group to work through the cortex in association with a cognitive process. Subsequently, different explanatory statistical models have been integrated into the exercises and associated cerebral cortex.

  1. The sample size was too low, and the inter-subject variability was very high. Therefore, the results of the study do not seem to be reliable.

R= We agree with you that the sample size is small, however, we believe that the results may be considered preliminary and that games such as Lego may be part of future clinical practice. Furthermore, considering other types of criteria for calculating sample size in analysis of covariance (ANCOVA) tests, we consider that the sample size used in this article may be within the appropriate minimum to guarantee a significant difference.

Borm et al. (2007) proposed a formula to compute the sample size for analysis of covariance (ANCOVA), and they concluded that an ANCOVA can considerably reduce the number of patients required for a clinical trial. For example, for the Orbitomedial Cortex, following the formula of Borm et al. (2007), we first compute the sample size using the t test formula to test the mean values of two treatments, considering a mean Pretest value of 59.8 for both groups, and according to the results of the ANCOVA, the Posttest values for Control 73.3 and for Lego 88.0, and standard error of 15.21; then, the sample size is n=18. The second step is multiplied by a factor of 2*(n+1)*(1-R2), where R2 is the R-squared of the linear regression (the square of the correlation coefficient R=8375); therefore, the formula of Borm et al. (2007) yields a minimum sample size of N=11. The results for the other features are given in the following table:

n using t test

N using ANCOVA

N = 2*(n+1)*(1-R2)

Attention and Executive Function

153

111

Memory

36

20

Attention and Memory

93

100

Orbitomedial Cortex

18

11

Anterior Prefrontal Cortex

12

12

Dorsolateral Cortex

12

8

Total Executive Functions

41

50

Note in the table that those variables where we find a significant difference, under the sample sizes that we have in the study, are those that coincide with these small sample sizes.

On the other hand, Shan and Ma (2014) compare the approach of Borm et al. (2007) with an exact approach, using the noncentral F distribution, for analysis of covariance with one covariate, with similar results.

Another more recent work on sample size is that carried out by Bujang et al. (2017) for linear regression model (LRM) and analysis of covariance (ANCOVA). Bujang et al. (2017) estimated the minimum sample size required for LRM and ANCOVA when R-squared was used as the effect size. Although they suggested that a sample size of 300 is necessary for clinical trials, they generated tables for the different R2 values for the control and treatment groups. For example, for the Orbitomedial Cortex with R2 of the Control and Lego groups given by R2_C=0.78 and R2_Lego=0.57, with a single variable for the group, a sample size of 10 would be enough to show significant differences.

Moreover, there are studies that have used interventions with LEGO therapy and have also recruited few patients (up to less than 10) to demonstrate the effect of the therapy on neuropathologies, for example:

  1. Expert Syst. 2015, 32(6), 698-709 (n=6 subjects).

2.Autism Dev Disord. 2015, 45(11), 3746-3755 (n=6 subjects)

3.JFMH. 2021, 23(5), 359-366. (n=30 subjects)

  1. JHET. 2020, 11(1), 24-31. (n=16 subjects)
  2. Biomed Mater Eng. 2014, 24(6), 3549-3556. (n=9 subjects)

6.Int J Environ Res Public Health. 2016, 13(12), 1176. (n=7 subjects)

  1. Psychol Stud. 2017, 62, 142-151 (n=2 subjects).

Finally, it should be noted that this study was carried out over more than two years of intense work and that, initially, 45 patients and their parents or guardians agreed to participate; however, the number of patients was reduced by almost half because the parents or guardians did not continue to attend all the sessions necessary to perform the neuropsychological evaluations and the LEGO therapy intervention, which was a criterion for exclusion. On the other hand, the patients who were ultimately enrolled strictly complied with the evaluations and all the indications for the intervention, in addition to the fact that they were evaluated by highly qualified and trained staff, so that any results obtained were real and reliable. We understand the importance of increasing the sample size, and we are working to better control the factors that contributed to patients dropping out of the study. Nevertheless, we also believe that this study is important and offers reliable data that can contribute to patients having more and better options in their neurohabilitation.

References

Barakova, E., Bajracharya, P., Willemsen, M., Lourens, T., & Huskens, B. (2014). Long‐term lego therapy with humanoid robot for children with asd. Expert Systems, 32(6), 698-709. https://doi.org/10.1111/exsy.12098

Bazoolnejad, M., Vakili, S., VAHID, L. K., & Yaripour, M. (2021). The effectiveness of lego therapy on the resiliency of gifted children. Journal of Fundamentals of Mental Health.

Borm G.F., Fransen J., Lemmens W.A.J.G. (2007). A simple sample size formula for analysis of covariance in randomized clinical trials. Journal of Clinical Epidemiology; 60, 1234-1238.

Bujang M.A., Sa’at N., Ikhwan T.M., Sidik T.A.B. (2017). Determination of Minimum Sample Size Requirement for Multiple Linear Regression and Analysis of Covariance Based on Experimental and Non-experimental Studies. Epidemiology Biostatistics and Public Health; 14(3):e12117.

Harn, P. (2017). A preliminary study of the empowerment effects of strength-based LEGO® SERIOUS PLAY® on two Taiwanese adult survivors by earlier domestic violence. Psychological studies, 62(2), 142-151.

Huskens, B., Palmen, A., Van der Werff, M., Lourens, T., & Barakova, E. (2015). Improving Collaborative Play Between Children with Autism Spectrum Disorders and Their Siblings: The Effectiveness of a Robot-Mediated Intervention Based on Lego® Therapy. Journal of autism and developmental disorders, 45(11), 3746–3755. https://doi.org/10.1007/s10803-014-2326-0.

Lopez-Samaniego, L., & Garcia-Zapirain, B. (2016). A Robot-Based Tool for Physical and Cognitive Rehabilitation of Elderly People Using Biofeedback. International journal of environmental research and public health, 13(12), 1176. https://doi.org/10.3390/ijerph13121176

Lopez-Samaniego, L., Garcia-Zapirain, B., & Mendez-Zorrilla, A. (2014). Memory and accurate processing brain rehabilitation for the elderly: LEGO robot and iPad case study. Bio-medical materials and engineering, 24(6), 3549–3556. https://doi.org/10.3233/BME-141181.

Shan G., Ma C. (2004). A Comment on Sample Size Calculation for Analysis of Covariance in Parallel Arm Studies. Biometrics & Biostatistics, 5:1.

Shields, M., Hunnell, W., Tucker, M., & Price, A. (2020). Building Blocks and Coloring Away Stress: Utilizing Lego® and Coloring as Stress Reduction Strategies among University Students. Journal of Health Education Teaching, 11(1), 24-31.

  1. Line 268. "The effect of therapy in the CTRL and LEGO® B-T patients..." What kind of therapy did the control group receive? Figures 3 and 4 demonstrate difference between the pretest and the posttest results in the CTRL group. As shown in Figure 4, the total executive function scores in the control group improved from the pretest to the posttest period (same to what is seen after LEGO®-based therapy). The similar dynamics of executive functions might be associated with the positive dynamics of epilepsy and the effectiveness of medical therapy in the control group.

R= Thank you for the observation. The CTRL group did not receive any therapy; this has been corrected in the manuscript on page 12. Although it is true that for the CTRL group the posttest also yielded better scores than the pretest (as in the LEGO® B-T group), particularly for total executive functions, and this improvement could be explained by the positive dynamics of epilepsy and the effectiveness of medical therapy in the CTRL group, figures 3 and 4 show that the LEGO® B-T group exhibited a statistically significant improvement and that this improvement tended to increase compared to that in the CTRL group. In addition, ANCOVA and Wilcoxon tests were used to evaluate the change between the posttest and the pretest, and these changes were compared between the CTRL and the LEGO® B-T groups.

  1. The initial results of the neuropsychological tests in the control group and the treated group were remarkably different (pretest values in Figures 3 and 4). The subjects in the treated group initially showed better attention and executive functions than CTRL group, as indicated by the results of the NEUROPSI test (Figure 3). They also demonstrated enhanced dorsolateral cortex functions in comparison to the CTRL group (Figure 4) and improved total executive functions, as measured by the BANFE-2 test (Figure 4). Therefore, the CTRL group in this study appears to be a distinct cohort.

R= The CTRL group is not from a different cohort. When incorporating the patients into the protocol, the intervention therapy was first explained to the parents of the BEFORE to apply any test, and they were invited to participate. The parents and their children were free to decide whether to participate. If they agreed, they were in the LEGO® B-T group, and if they did not want to, they were only given the neuropsychological tests once and six months later. Moreover, although the selection of patients was not completely random, the study was blinded. We add this information on page 3.

  1. The authors concluded (page 14) that "cognitive habilitation of pediatric epilepsy patients using LEGO® B-T caused significant changes in the OMC, APF and DL areas and in memory". The aforementioned cortical areas were not examined in this study. The authors assessed the functions of these areas, but not the changes.

R= You are correct. The manuscript has been corrected on page 16.

Minor.

The legends for Figures 3 and 4 are redundant in the text.

R=Thank you for your comment. The text in section 3.4 has been corrected on page 12 and 13.

Figure 2 illustrates the marked variability in test executive scores between subjects. The top and bottom plots are the same, except for some differences in color. The authors are encouraged to consider a more effective illustration of the fact that "a patient with structural etiology and combined epilepsy and a patient with genetic etiology and focal epilepsy showed the lowest performance" (lines 282-284).

R=Thank you for your comment. We have changed the graphs, and we have rewritten the text focusing on the factors that explain the variability of execution that occurs during the sessions, page 11.

Figure 1 and Table 3 present the same data. Figure 1 illustrates the underlying patterns and trends that appear to be quite similar in the CRTL and the treated group.

R= While it is true that the data in figure 1 comes from table 3 data, figure 1 (as you mention) shows the underlying patterns and trends of the subjects' data. These trends were subsequently verified through statistical tests, and although they were apparently similar between the CTRL and experimental groups, the statistics showed significant improvement in the LEGO® B-T group and a trend toward better development compared to the CTRL group (Figures 3, 4 and 5).

Reviewer 2 Report

Comments and Suggestions for Authors

The research topic is relevant and interesting. Cognitive disorders in children and adolescents with epilepsy can be caused by various causes, including the frequency and severity of seizures, the etiology of childhood epilepsy (structural, metabolic, infectious, autoimmune, genetic), undesirable drug reactions with prolonged use of antiepileptic drugs, especially in maximum daily doses.

In this regard, I recommend modifying the Introduction section and focusing potential readers' attention specifically on the problem of cognitive disorders associated with epilepsy in children and adolescents, including the frequency, causes, and possible ways of drug and non-drug correction.

At the end of the Introduction section, add a clearly stated purpose for your research.

In the Materials and Methods section, explain how the sample size was calculated to eliminate systematic errors during subsequent statistical processing of the database. The main reason for my doubts is that the sample size is very small - 22 participants, including two comparison groups (12 and 10 participants in the main and control groups, respectively). This is very small and does not correspond to the existing approaches to the formation of the sample size (for example, according to the Altman nomogram). Taking into account the known indicators of the frequency of cognitive disorders in pediatric epilepsy, the estimated number in each group should be at least 30.

Specify the minimum and maximum age of the study participants in the Inclusion Criteria.

Subsections 2.3.1 and 2.3.2: Please add units of measurement (points?).

Check all abbreviations, they should be explained at the first use.

In the Materials and Methods section, it is necessary to add detailed information about which antiepileptic drugs and in what doses were prescribed to the study participants.

In subsection 3.1, the authors write that the patients had structural and unspecified epilepsy, but table 2 also lists cases of genetic epilepsy.  Thus, this is not only a very small sample, but a very heterogeneous one. The statistical analysis of the data also raises a question. In particular, it is not correct to calculate the proportion of the number of cases as a percentage of 12 and 10.

In general, the results of this study are highly questionable, despite a good idea and a new approach to choosing non-drug (gaming) correction of cognitive disorders in childhood epilepsy. The sample size needs to be increased.

Author Response

Responses to reviewer 2.

The research topic is relevant and interesting. Cognitive disorders in children and adolescents with epilepsy can be caused by various causes, including the frequency and severity of seizures, the etiology of childhood epilepsy (structural, metabolic, infectious, autoimmune, genetic), undesirable drug reactions with prolonged use of antiepileptic drugs, especially in maximum daily doses.

  1. In this regard, I recommend modifying the Introduction section and focusing potential readers' attention specifically on the problem of cognitive disorders associated with epilepsy in children and adolescents, including the frequency, causes, and possible ways of drug and non-drug correction.

R=Thank you for your comment. The introduction has been modified.

  1. At the end of the Introduction section, add a clearly stated purpose for your research.

R= The aim of the work was modified (page 3).

  1. In the Materials and Methods section, explain how the sample size was calculated to eliminate systematic errors during subsequent statistical processing of the database. The main reason for my doubts is that the sample size is very small - 22 participants, including two comparison groups (12 and 10 participants in the main and control groups, respectively). This is very small and does not correspond to the existing approaches to the formation of the sample size (for example, according to the Altman nomogram). Taking into account the known indicators of the frequency of cognitive disorders in pediatric epilepsy, the estimated number in each group should be at least 30.

R= We agree with you that under the Altman nomogram criterion [Altman, 1980], to compute the minimum sample size to test the difference in the means of two treatments, a minimum sample size of 30 is required to guarantee a statistically significant difference, considering a significance level of alpha=0.05 (type I error size of 0.05) and a power of 0.80 (type II error size of 0.20), given that for the observed data, the minimum standardized difference is 1.0. However, we believe that the results may be considered preliminary and that games such as Lego may be part of future clinical practice. Furthermore, considering other types of criteria for calculating sample size in analysis of covariance (ANCOVA) tests, we consider that the sample size used in this article may be within the appropriate minimum to guarantee a significant difference.

Borm et al. (2007) proposed a formula to compute the sample size for analysis of covariance (ANCOVA), and they concluded that an ANCOVA can considerably reduce the number of patients required for a clinical trial. For example, for the Orbitomedial Cortex, following the formula of Borm et al. (2007), we first compute the sample size using the t test formula to test the mean values of two treatments, considering a mean Pretest value of 59.8 for both groups, and according to the results of the ANCOVA, the Posttest values for Control 73.3 and for Lego 88.0, and standard error of 15.21; then, the sample size is n=18. The second step is multiplied by a factor of 2*(n+1)*(1-R2), where R2 is the R-squared of the linear regression (the square of the correlation coefficient R=8375); therefore, the formula of Borm et al. (2007) yields a minimum sample size of N=11. The results for the other features are given in the following table:

n using t test

N using ANCOVA

N = 2*(n+1)*(1-R2)

Attention and Executive Function

153

111

Memory

36

20

Attention and Memory

93

100

Orbitomedial Cortex

18

11

Anterior Prefrontal Cortex

12

12

Dorsolateral Cortex

12

8

Total Executive Functions

41

50

Note in the table that those variables where we find a significant difference, under the sample sizes that we have in the study, are those that coincide with these small sample sizes.

In addition, Shan and Ma (2014) compare the approach of Borm et al. (2007) with an exact approach, using the noncentral F distribution, for analysis of covariance with one covariate, with similar results.

Another more recent work on sample size is that carried out by Bujang et al. (2017) for linear regression model (LRM) and analysis of covariance (ANCOVA). Bujang et al. (2017) estimated the minimum sample size required for LRM and ANCOVA when R-squared was used as the effect size. Although they suggested that a sample size of 300 is necessary for clinical trials, they generated tables for the different R2 values for the control and treatment groups. For example, for the Orbitomedial Cortex with R2 of the Control and Lego groups given by R2_C=0.78 and R2_Lego=0.57, with a single variable for the group, a sample size of 10 would be enough to show significant differences.

On the other hand, there are studies that have used interventions with LEGO therapy and have also recruited few patients (up to less than 10) to demonstrate the effect of the therapy on neuropathologies, for example:

  1. Expert Syst. 2015, 32(6), 698-709 (n=6 subjects).

2.Autism Dev Disord. 2015, 45(11), 3746-3755 (n=6 subjects)

3.JFMH. 2021, 23(5), 359-366. (n=30 subjects)

  1. JHET. 2020, 11(1), 24-31. (n=16 subjects)
  2. Biomed Mater Eng. 2014, 24(6), 3549-3556. (n=9 subjects)

6.Int J Environ Res Public Health. 2016, 13(12), 1176. (n=7 subjects)

  1. Psychol Stud. 2017, 62, 142-151 (n=2 subjects).

Moreover, it should be noted that this study was carried out over more than two years of intense work and that, initially, 45 patients and their parents or guardians agreed to participate; however, the number of patients was reduced by almost half because the parents or guardians did not continue to attend all the sessions necessary to perform the neuropsychological evaluations and the LEGO therapy intervention, which was a criterion for exclusion. On the other hand, the patients who were ultimately enrolled strictly complied with the evaluations and all the indications for the intervention, in addition to the fact that they were evaluated by highly qualified and trained staff, so that any results obtained were real and reliable.

Finally, we understand the importance of increasing the sample size, and we are working to better control the factors that contributed to patients dropping out of the study. However, we also believe that this study is important and offers reliable data that can contribute to patients having more and better options in their neurohabilitation.

References

Altman D. (1980). Statistics and ethics in medical research: III How large a sample? British Medical Journal; 281: 1336-1338.

Barakova, E., Bajracharya, P., Willemsen, M., Lourens, T., & Huskens, B. (2014). Long‐term lego therapy with humanoid robot for children with asd. Expert Systems, 32(6), 698-709. https://doi.org/10.1111/exsy.12098

Bazoolnejad, M., Vakili, S., VAHID, L. K., & Yaripour, M. (2021). The effectiveness of lego therapy on the resiliency of gifted children. Journal of Fundamentals of Mental Health.

Borm G.F., Fransen J., Lemmens W.A.J.G. (2007). A simple sample size formula for analysis of covariance in randomized clinical trials. Journal of Clinical Epidemiology; 60, 1234-1238.

Bujang M.A., Sa’at N., Ikhwan T.M., Sidik T.A.B. (2017). Determination of Minimum Sample Size Requirement for Multiple Linear Regression and Analysis of Covariance Based on Experimental and Non-experimental Studies. Epidemiology Biostatistics and Public Health; 14(3):e12117.

Harn, P. (2017). A preliminary study of the empowerment effects of strength-based LEGO® SERIOUS PLAY® on two Taiwanese adult survivors by earlier domestic violence. Psychological studies, 62(2), 142-151.

Huskens, B., Palmen, A., Van der Werff, M., Lourens, T., & Barakova, E. (2015). Improving Collaborative Play Between Children with Autism Spectrum Disorders and Their Siblings: The Effectiveness of a Robot-Mediated Intervention Based on Lego® Therapy. Journal of autism and developmental disorders, 45(11), 3746–3755. https://doi.org/10.1007/s10803-014-2326-0.

Lopez-Samaniego, L., & Garcia-Zapirain, B. (2016). A Robot-Based Tool for Physical and Cognitive Rehabilitation of Elderly People Using Biofeedback. International journal of environmental research and public health, 13(12), 1176. https://doi.org/10.3390/ijerph13121176

Lopez-Samaniego, L., Garcia-Zapirain, B., & Mendez-Zorrilla, A. (2014). Memory and accurate processing brain rehabilitation for the elderly: LEGO robot and iPad case study. Bio-medical materials and engineering, 24(6), 3549–3556. https://doi.org/10.3233/BME-141181.

Shan G., Ma C. (2004). A Comment on Sample Size Calculation for Analysis of Covariance in Parallel Arm Studies. Biometrics & Biostatistics, 5:1.

Shields, M., Hunnell, W., Tucker, M., & Price, A. (2020). Building Blocks and Coloring Away Stress: Utilizing Lego® and Coloring as Stress Reduction Strategies among University Students. Journal of Health Education Teaching, 11(1), 24-31.

  1. Specify the minimum and maximum age of the study participants in the Inclusion Criteria.

R= Thank you for your comment. We added this information to the inclusion criteria on page 3. In addition, we specify the minimum and maximum ages of the participants in both groups in the Results section, page 6.

  1. Subsections 2.3.1 and 2.3.2: Please add units of measurement (points? ).

R= Yes, the units of measurement are natural points. We have added this to the mentioned sections on page 4.

  1. Check all abbreviations, they should be explained at the first use.

R= Abbreviations have been carefully reviewed and corrected.

  1. In the Materials and Methods section, it is necessary to add detailed information about which antiepileptic drugs and in what doses were prescribed to the study participants.

R=The information required has been incorporated into Table 3, page 8.

  1. In subsection 3.1, the authors write that the patients had structural and unspecified epilepsy, but table 2 also lists cases of genetic epilepsy. Thus, this is not only a very small sample, but a very heterogeneous one. The statistical analysis of the data also raises a question. In particular, it is not correct to calculate the proportion of the number of cases as a percentage of 12 and 10.

R= We apologize, but we do not clearly understand what you are requesting. Are you referring to putting the percentages in Table 2 with respect to 22 patients? If so, we consider that both ways of doing so are correct. However, if you recommend it, we could change the percentage based on the 22 patients.

  1. In general, the results of this study are highly questionable, despite a good idea and a new approach to choosing non-drug (gaming) correction of cognitive disorders in childhood epilepsy. The sample size needs to be increased.

R= In question 3, we broadly address this topic.

Reviewer 3 Report

Comments and Suggestions for Authors

LEGO-based therapy in pediatric epilepsy patiens could be very useful as indicated by  the introduction of the paper.

The study design is well documented and supported

The number of the patients is very low; the control group modulate this limitation, but the results can be considered as pleriminary data.

The results are clearly presented togheter with the tables and the figures.

The discussion elucidate all the significant changes that could be obtained through LEGO-based therapy.

On these basis could be very interesting to outline a future perspective 

though a clinical practice implementation program

Author Response

Responses to reviewer 3.

LEGO-based therapy in pediatric epilepsy patients could be very useful as indicated by the introduction of the paper.

R= Thank you for your comment.

The study design is well documented and supported

R= Thank you; we appreciate your comment.

The number of the patients is very low; the control group modulate this limitation, but the results can be considered as preliminary data.

R= Thank you; we agree with you.

The results are clearly presented together with the tables and the figures.

R= Thank you for your comment.

The discussion elucidates all the significant changes that could be obtained through LEGO-based therapy.

R= Thank you; we appreciate your comment.

On these bases could be very interesting to outline a future perspective though a clinical practice implementation program.

R= Thank you; the perspectives of the work were added on page 16.

Round 2

Reviewer 1 Report

Comments and Suggestions for Authors

I appreciate the author's thorough and detailed response to my critical questions. The manuscript has been updated in accordance with my critical remarks.
I have one concern that could lead to misunderstanding, which refers to issue 3 in my previous review. "The effect of therapy in ..."  I assumed that therapy was equivalent to pharmacotherapy, but the authors assumed the effect of LEGO-based therapy instead. To avoid confusion, I propose replacing "therapy" with "LEGO-based therapy" in the captions of Figures 3 and 4.

Thank you for adding Table 3, which includes a list of the AEDs used for treating epilepsy in each patient.
The results of cognitive tests in the control group could be linked to the effect of pharmacotherapy. In contrast, the treated group showed a combined effect of both pharmacotherapy and LEGO-based therapy. In order to assess the effect of LEGO-based therapy, we have to assume that the pharmacotherapy was equally effective in both groups of patients. As demonstrated in Figure 4, the control group demonstrated an improvement in their total executive function scores from the pretest to the posttest period. Therefore, positive dynamics of executive functions might be associated with the effectiveness of pharmacotherapy in the control group. The treated group might show a greater effectiveness in pharmacotherapy compared to the control group. Could the authors please comment on this issue?

Author Response

Responses to reviewer 1.

I appreciate the author's thorough and detailed response to my critical questions. The manuscript has been updated in accordance with my critical remarks

R=We also appreciate your comments, because we believe that the manuscript has been significantly improved.

I have one concern that could lead to misunderstanding, which refers to issue 3 in my previous review. "The effect of therapy in ..."  I assumed that therapy was equivalent to pharmacotherapy, but the authors assumed the effect of LEGO-based therapy instead. To avoid confusion, I propose replacing "therapy" with "LEGO-based therapy" in the captions of Figures 3 and 4.

R= The captions of Figures 3, 4 and 5 have been modified. In addition to some “therapy” words in the manuscript have been modified

Thank you for adding Table 3, which includes a list of the AEDs used for treating epilepsy in each patient.

R=Thank you for your suggestion, we also believe that the paper has been improved with that information.

The results of cognitive tests in the control group could be linked to the effect of pharmacotherapy. In contrast, the treated group showed a combined effect of both pharmacotherapy and LEGO-based therapy. In order to assess the effect of LEGO-based therapy, we have to assume that the pharmacotherapy was equally effective in both groups of patients. As demonstrated in Figure 4, the control group demonstrated an improvement in their total executive function scores from the pretest to the posttest period. Therefore, positive dynamics of executive functions might be associated with the effectiveness of pharmacotherapy in the control group. The treated group might show a greater effectiveness in pharmacotherapy compared to the control group. Could the authors please comment on this issue?

R=Thank you by your comment, in the discussion we added information related to the effect of the antiepileptic drugs in cognition, page 16.

Reviewer 2 Report

Comments and Suggestions for Authors

The authors have modified the manuscript and responded to my comments. The quality of the manuscript has been improved, but it still needs a minor technical revision.

I recommend adding the authors' explanations of the small sample size to the separate Limitation section (after the Discussion section). This will improve the readers' interest in this article. Add the appropriate links to the References.

Line 291: Remove the abbreviation “AEDs” from the table name, move the abbreviation and its explanation to the note after the table.

Line 294: Remove the abbreviations “CTRL” and “LEGO® B-T” from the table name, move the abbreviation and its explanation to the note after the table. Add the name of the first column (Patient?). Add a Note under table 4 and explain all the abbreviations used in this table.

Add a Note to all figures and explain the abbreviations used.

Author Response

Responses to reviewer 2.

The authors have modified the manuscript and responded to my comments. The quality of the manuscript has been improved, but it still needs a minor technical revision.

I recommend adding the authors' explanations of the small sample size to the separate Limitation section (after the Discussion section). This will improve the readers' interest in this article. Add the appropriate links to the References.

R= Thank you for your recommendation. The limitation section (page 17) and references have been added.

 Line 291: Remove the abbreviation “AEDs” from the table name, move the abbreviation and its explanation to the note after the table.

R= The abbreviation has been modified.

Line 294: Remove the abbreviations “CTRL” and “LEGO® B-T” from the table name, move the abbreviation and its explanation to the note after the table. Add the name of the first column (Patient?). Add a Note under table 4 and explain all the abbreviations used in this table.

R= The information required has been modified.

Add a Note to all figures and explain the abbreviations used.

R= Thank you for your comment. The information required has been modified.

Round 3

Reviewer 2 Report

Comments and Suggestions for Authors

I thank the authors for their attention to my comments and recommendations.

The manuscript has been re-modified by the authors and can be accepted for publication.